# The Molecular Basis of Asthma Exacerbations Triggered by Viral Infections: The Role of Specific miRNAs

**DOI:** 10.3390/ijms26010120

**Published:** 2024-12-26

**Authors:** Natalia Kierbiedź-Guzik, Barbara Sozańska

**Affiliations:** Department and Clinic of Paediatrics, Allergology and Cardiology, Wroclaw Medical University, ul. Chałubińskiego 2a, 50-368 Wrocław, Poland

**Keywords:** miRNA, biomarkers, asthma, exacerbation, children, rhinovirus, respiratory syncytial virus, influenza virus, metapneumovirus, Severe Acute Respiratory Syndrome Coronavirus 2

## Abstract

Viral respiratory infections are a significant clinical problem among the pediatric population and are one of the leading causes of hospitalization. Most often, upper respiratory tract infections are self-limiting. Still, those that involve the lower respiratory tract are usually associated with asthma exacerbations, leading to worsening or even the initiation of the disease. A key role in regulating the immune response and inflammation during viral infections and their impact on the progression of asthma has been demonstrated for miRNA molecules (microRNA). Their interaction with mRNA (messenger RNA) regulates gene expression in innate and acquired immune responses, making them valuable biomarkers for diagnostics, monitoring, and predicting asthma exacerbations. The following paper presents changes in the expression of miRNAs during the five most common viral infections causing asthma worsening, with particular emphasis on the pediatric population. In addition, we describe the molecular mechanisms by which miRNAs influence the pathogenesis of viral infection, immune responses, and asthma exacerbations. These molecules represent promising targets for future innovative therapeutic strategies, paving the way for developing personalized medicine for patients with viral-induced asthma exacerbations.

## 1. Introduction

Respiratory tract infections are the most common diseases occurring in the pediatric population [1] and represent a significant global health problem, substantially contributing to increased hospitalizations and death [2,3]. Viruses such as rhinoviruses and respiratory syncytial viruses dominate the etiological factors [4,5]. Identifying specific infections is reliable through widely accessible diagnostic methods for respiratory diseases, such as PCR tests. Regarding location differences, these illnesses can be categorized into those that impact the upper and lower respiratory tract [6,7]. In settings, most instances involve the upper respiratory system (throat, tonsils, nasopharynx, middle ear, and paranasal sinuses) and typically resolve spontaneously. There are also cases where infections target the larynx, trachea, bronchi, and lungs [8]. In infancy, viral infections often trigger recurring wheeziness, which may result in asthma later in life [9]. Respiratory infections play a role in causing asthma flare-ups in both adults and children [10,11].

miRNA are non-coding RNA fragments, 18–22 nucleotides in length, that, by interacting with mRNA, cause its degradation or inhibit its translation, influencing various inflammatory, immunological, and immune processes [12]. They are essential for the proper development of the respiratory tract and lungs. Throughout childhood, they aid in precisely tuning respiratory inflammatory processes, including respiratory infections. In asthma patients, miRNAs play roles in helping with diagnosis and monitoring disease progression, evaluating treatment response, and predicting the likelihood of future disease flare-ups. They have a hand in controlling all functions, but our understanding of their precise workings is still unclear and needs to be made more transparent, particularly regarding viral infections. How miRNAs are expressed in cells is significantly involved in how viruses interact with their hosts, including steps like entering cells to replicate and spread. Additionally, many of them impact gene expression regulation and modulation, which is critical to the pathogenesis of various viruses [13,14].

This review comprehensively analyzes miRNA expression variability about the most common viral respiratory infections and their impact on asthma progression. Drawing on extensive clinical and experimental data, it delves into the molecular mechanisms by which specific miRNAs may aggravate symptoms and contribute to asthma flare-ups. Insights from adult and pediatric studies and cellular models are integrated to present a complete overview of miRNA expression changes during viral infections.

While existing research has extensively explored miRNA dynamics in infections, this work centers on differential miRNA expression in the five viral infections most frequently linked to asthma exacerbations in children. Future investigations may extend these findings to include additional pathogens. The insights gained here will lay the groundwork for developing miRNA-targeted therapeutic approaches.

## 2. Asthma and Viral Infection

Undoubtedly, asthma is the most common chronic inflammatory condition affecting children’s respiratory systems globally [15]. The typical signs of this condition encompass coughs, sudden breathlessness, and wheezes. A range of elements, such as the level of disease control and adherence to advice, play a role in triggering the worsening of this ailment in children. Viral infections are marked as one of the factors impacting the seriousness of asthma. They heighten the chances of this disorder’s occurrence and contribute significantly to its deterioration during the fall or winter months. This leads to frequent visits to the hospital emergency department and higher rates of hospitalization along with increased healthcare expenses and, in severe instances, even fatalities [16].

In viral infections, interferons (IFNs) are critical components involved in the immune response to viral infections. Their induction in patients with asthma is usually delayed and impaired, which results in greater susceptibility to viral infections and severity of symptoms. There is information in the literature that a low IFN type I response at the beginning of the disease increases viral replication and predicts the risk of short-term adverse events. Moreover, among children with asthma, it has been noted that a low IFN response in the later stage of the disease may result in its severity, which may additionally lead to a deterioration of health. miRNAs also participate in the interferon response. They can either suppress or support the expression of genes associated with the synthesis of this compound, which affects the strength of the antiviral response [17].

Research has shown a distinct expression profile of microRNAs in different tissues and bodily fluids in asthmatic patients and healthy individuals. These differences could serve as potential diagnostic markers for early asthma detection and predicting the risk of exacerbations. One example is the molecule miRNA-21, whose levels are increased in the bodily fluids of patients with confirmed asthma [18]. This molecule exacerbates inflammation and bronchial hyperreactivity and leads to lung structural changes. Furthermore, its levels are often elevated in many cases of viral infections. Despite its role in activating T cells and stimulating the immune system, viral replication frequently increases in many infections. This situation further elevates miRNA-21 expression in individuals with diagnosed asthma and confirmed viral infection. Ultimately, through known molecular pathways, this can lead to exacerbation of the underlying disease [19].

Viruses can disrupt miRNA biogenesis by directly competing with cellular pre-miRNA or indirectly affecting their expression due to the host’s immune response to the foreign agent. Numerous viral infections have been associated with abnormal expression of various host miRNAs, allowing viruses to upregulate or downregulate host miRNAs to evade the immune system. Changes in the host’s miRNA expression profile induced by viral infection participate in various signaling pathways (Figure 1), modulating host–virus interactions and regulating viral infectivity and activation of the immune response [20,21].

In the following sections of the article, we will present how specific viruses affect the alteration of miRNA and the molecular targets of these molecules. The pathogens in the paragraph are listed in order of those most frequently causing asthma exacerbations in children, concluding with SARS-CoV-2, a novel pathogen that may also significantly worsen the disease.

## 3. Rhinovirus (RV)

One of the primary defense mechanisms against pathogens is the innate immune response, which involves various cells such as dendritic cells, monocytes, macrophages, granulocytes, and natural killer (NK) cells. Infections can alter the expression of specific miRNA molecules, impairing immune function and consequently affecting the progression of many respiratory diseases, including asthma [22]. Rhinovirus is a non-enveloped RNA virus responsible for over 50% of human upper respiratory tract infections, where three species can be distinguished: RV-A, RV-B, and RV-C [23]. RV is a primary etiological factor alongside RSV in causing bronchiolitis in infants and asthma development. In the first months of life, these infections can trigger a Th2-dependent response (T helper 2), negatively impacting lung development and increasing the risk of chronic lung disease [24]. Additionally, RV and allergens promote increased production of IL-33 and cytokines, critical mediators of inflammation and airway remodeling, leading to asthma exacerbations in childhood. One identified mechanism for worsening symptoms was the increased type III interferon production [25,26].

Two of the molecules extensively studied in the context of rhinovirus infection were miRNA-146a and miRNA-146b. The experiment used human bronchial epithelial cells and mice in which the disease was induced. The expression of these molecules increased after exposure to the pathogen. However, when miRNA-146a was administered intranasally before infection, there was a reduction in airway inflammation. They work by reducing the production of chemokines that attract neutrophils and stimulating the production of interferon λ, which reduces viral spread. These specific molecules may reduce inflammation by blocking the signaling pathway NF-κB (nuclear factor kappa B), which influences the expression of various cytokines. When levels of miRNAs 146a/146b were reduced, there was an increase in the number of neutrophils entering the affected areas. This finding suggests a link between miRNA-146a levels and the development of asthma, which is mainly driven by neutrophils. Decreased levels of these molecules in allergic airway inflammation, mainly caused by house dust mites, worsened inflammation after rhinovirus exposure and caused a weaker Th2 immune response. This may be due to the release of cytokines associated with Th1 and Th17 responses. MiRNAs 146a/146b are crucial in controlling the reaction to rhinoviruses. It is essential to ensure their expression to treat airway inflammation in people with asthma [27,28,29].

Another interesting molecule that plays a vital role in viral infections is miRNA-155. Among children admitted to the hospital due to respiratory tract infections, a PCR nasopharyngeal swab was performed to determine the etiology of the disease. Patients of appropriate age were recruited into the control and study groups. The concentration of this molecule in the collected material was higher among infected patients, and its increased expression was associated with reduced severity of respiratory disease and an increased Th1 antiviral response. Additionally, increased levels of proinflammatory cytokines characteristic of the Th1-mediated response were noted among these subjects. Besides the molecule’s participation in the antiviral response, it also significantly affects the development of various CD4+ T cell subtypes, such as Treg, Th2, and Th17 [30,31,32]. When Th2 cells become active, the expression of miRNA-155, stimulated by the host immune system, increases and inhibits the function of sphingosine-1-phosphate receptor-1 (S1pr1) [33]. This action leads to the movement of Th2 helper cells toward the lungs, potentially influencing the onset of asthma and how the immune system responds to infections. The results of another study led to interesting conclusions. Human bronchial epithelial cell lines were used for the experiment. AntimiRNA molecules were employed to inhibit the expression of specific miRNAs in the cells, and the DICER enzyme activity was suppressed. Then, rhinovirus was introduced into the material. This demonstrated increased replication of the pathogen, proving the critical importance of these molecules for the virus. Subsequently, based on previous reports, molecules underexpressed in asthmatic bronchial epithelial cells and predicted to target different rhinovirus strains were selected for further study. It has been shown that the lack of miRNA-155 and miRNA-128 among asthma patients makes them particularly susceptible to worsening symptoms during rhinovirus infection. These results underline the role of miRNAs in modulating the immune defense against respiratory viruses [34,35].

A study investigating the impact of two specific miRNA molecules on rhinovirus infection identified significant changes in miRNA levels over time. Among the nine miRNAs affected, miRNA-101-3p and miRNA-30b-5p showed the strongest association with genes involved in the immune response. MiRNA-101-3p interacts with the *EZH2* (Enhancer of Zeste 2 Polycomb Repressive Complex 2), a gene crucial for maintaining immune balance. A reduction in EZH2 can weaken antiviral defenses. Meanwhile, miRNA-30b-5p targets *RARG* (Retinoic Acid Receptor Gamma) and *PTPN13* (Protein Tyrosine Phosphatase, Non-Receptor Type 13), inhibiting the expression of proteins coded by these genes. In asthma patients, elevated levels of miRNA-30b-5p are linked to activation of the PI3K/AKT (Phosphoinositide 3-Kinase/Protein Kinase B) pathway, contributing to airway smooth muscle dysfunction and remodeling [36]. In addition, reducing *RARG* expression could reduce the host response to Staphylococcus aureus. Studies suggest that cells infected with rhinoviruses increase S. aureus after initial bacterial clearance due to problems with various cellular structures. This interesting finding is of great importance, especially among patients with asthma, because rhinovirus infection, which causes an exacerbation of airway inflammation, can worsen this condition by increasing the replication of S. aureus. The consequence of all these events is the worsening of disease symptoms. This observation is an interesting direction for further research on the interactions between viruses and bacteria [37,38].

The following study recruited patients with mild allergic asthma and healthy participants for an observational, longitudinal, two-phase case-control study. Participants were assessed at baseline and then after RV16 vaccination. Blood samples were collected before and after the onset of viral infection. Two groups of miRNA molecules were isolated from the serum of asthmatic patients, with levels assessed at baseline and several days later. The first group of miRNAs showed increasing expression in response to infection, associated with Th1 responses and interferon-stimulated cytokines, such as interferon λ and IP-10. These molecules play a crucial role in antiviral and anti-inflammatory processes. However, patients exhibited impaired responses to IFN λ, potentially affecting asthma severity and increasing the risk of other infections. MiRNA-21-5p and miRNA-126-3p interacted with the *RSAD2* gene (Radical S-adenosyl methionine domain containing 2), which is significant for the antiviral response to rhinovirus. In the second group, a decreasing trend in serum levels of miRNAs was observed, correlating strongly with Th2 responses. Cytokines produced by Th2 cells, such as interleukin-4 and interleukin-13, may inhibit interferon production against rhinovirus. Both miRNA groups targeted critical genes involved in the antiviral response. This study highlights potential therapeutic targets for preventing rhinovirus-induced asthma development and exacerbation [39,40,41].

The next study recruited middle-aged and older asthmatics who were admitted to the outpatient clinic for asthma exacerbation and then for follow-up 6–8 weeks after the event. During these two visits, blood samples were collected from the subjects, and peripheral blood mononuclear cells (PBMCs) were isolated from them and cultured with rhinovirus 1b or control medium. The expression of miRNA-19b, -106a, -126a, and -146a was assessed in serum and PBMCs by RT-PCR, while cytokine levels (INF-γ, TNF-α, IL-6, and IL-10) were analyzed in culture supernatants using flow cytometry. The results showed higher expression of miRNA-126a and miRNA-146a in serum during asthma exacerbations compared with the follow-up visit. However, no significant changes in miRNA expression were detected at any time point after PBMC exposure to rhinovirus despite increased cytokine production, indicating an enhanced immune response despite stable miRNA levels. This suggests that cytokine activity, rather than changes in miRNA, may play a key role during asthma exacerbations [42].

## 4. Respiratory Syncytial Virus (RSV)

Respiratory syncytial virus is a prevalent pathogen among infants, responsible for about 80% of lower respiratory tract infections, particularly around 2–3 months of age [43]. It is the leading trigger of bronchiolitis and pneumonia in young children and increases the risk of recurrent wheezing and asthma in 30–40% of cases [14]. RSV is the top cause of hospitalizations in children under 12 months, and globally, it ranks second after malaria as a leading cause of infant mortality. In temperate regions, RSV infections surge in late fall through spring [44,45].

Among children hospitalized for bronchiolitis, 16 patients with confirmed RV or RSV infection alone were randomly included in a multicenter study. Nasopharyngeal swabs were collected within the first 24 h. Then, the expression profile of miRNA, mRNA, and cytokines was analyzed using specialized methods. Among RV-infected children, an increase in miRNA targeting the *NFKBIB* gene (Nuclear Factor of Kappa Light Polypeptide Gene Enhancer in B-Cells Inhibitor Beta) was observed, inhibiting its expression and reducing the production of kappaB inhibitors. This activated the NF-κB pathway, which is responsible for airway hyperreactivity, allergic inflammation, and elevated type 2 cytokine production. miRNA-155, significant in pro-asthmatic responses, was elevated. In contrast, RSV-infected infants showed a different miRNA profile with reduced NF-κB activity. This experiment supports the notion that RV infection increases the likelihood of asthma development compared to RSV, providing molecular evidence [46,47,48,49].

The previous part of the article highlighted the role of miRNA-146a as an essential molecule in the progression of RV infection, which also has implications for RSV. In the study of Huang et al., which examined two different human cell lines and rats, it was noted that RSV exposure resulted in a decrease in miRNA-146a levels in all biological samples, in contrast to previous results indicating an increase during inflammatory reactions. Furthermore, the application of a miRNA-146a mimic before pathogen exposure resulted in attenuation of inflammation by reducing the production of IL-1β, IL-6, and TNF-α, as well as inhibition of TRAF-6 (TNF receptor-associated factor 6) and JNK (Jun N-terminal kinase)/ERK (extracellular signal-regulated kinase)/NF-κB/MAPK (mitogen-activated protein kinase) pathways involved in the regulation of inflammation [50]. Another study focused on the role of miRNA-146a in the context of RSV infection, recruiting infants with confirmed infections through nasopharyngeal swabs. This study, like previous ones, confirmed reduced expression of miRNA-146a in biological samples. In children who had experienced bronchiolitis and continued breathing difficulties, miRNA-146a was lower than in healthy children but higher than in those with an active infection. This suggests that low levels of this molecule may be associated with prolonged inflammation lasting even after recovery from the illness. Utilizing this information could aid in identifying infants at risk of persistent inflammation and respiratory issues [51].

Increased mucus production is a key mechanism contributing to severe bronchitis and heightened asthma exacerbation risk due to RSV exposure. This condition leads to airway hyperreactivity and increases the likelihood of childhood asthma development [52]. The involvement of miRNA-34b/34c has been confirmed, with reduced expression observed in RSV-infected individuals and cell cultures. Their action is linked to the stimulation of FGFR1 (Fibroblast Growth Factor Receptor 1), which enhances MUC5AC (Mucin 5AC) production. Mimetics for these miRNAs reverse this process, indicating new therapeutic targets for RSV infection management [53]. The following two molecules, MiRNA-34b and miRNA-34c-5p, are notably expressed at reduced levels in individuals with asthma and COPD (chronic obstructive pulmonary disease) and in the lungs of RSV-exposed mice. These microRNAs primarily target CXCL10 (CXC Motif Chemokine Ligand 10), crucial in activating mast cells. This activation facilitates mast cell migration to airway smooth muscle, potentially worsening inflammation and bronchoconstriction associated with asthma. Furthermore, there is an observed increase in macrophage chemotaxis, highlighting an additional mechanism that may worsen existing respiratory conditions during RSV infection [54].

Advanced analytical methods revealed a connection between miRNA-27 and miRNA-26 with the infection in lung epithelial cells, both exposed and unexposed to RSV. MiRNA-27 regulates and inhibits the cell cycle and exhibits antiviral properties. MiRNA-26, in addition to its role in the immune response, affects long non-coding RNAs (lnRNAs), emphasizing its regulatory function within the nucleus. These findings underscore the complex role of miRNAs in viral infections, making their complete understanding essential for unraveling the progression of the disease [55].

The project recruited infants admitted to the emergency department due to respiratory tract infections. RSV infection was then confirmed among these subjects using rapid antigen tests or RT-PCR. Children with other chronic respiratory tract diseases were excluded from participation. The control group consisted of healthy infants recruited during routine visits to a family doctor. Then, using a modified mRDAI (Modified Respiratory Distress Assessment Instrument) tool, infants were classified as having a mild, moderate, or severe disease course. Nasopharyngeal aspirate samples and nasal epithelial cells were obtained. Differences in miRNA expression profiles were demonstrated between the study and control groups. Reduced expression was observed for miRNA-34b, miRNA-34c, miRNA-125b, miRNA-29c, miRNA-125a, miRNA-429, and miRNA-27b, whereas miRNA-155, miRNA-31, miRNA-203a, miRNA-16, and let-7d showed increased expression among infected patients. Notably, lower levels of miRNA-429 and miRNA-125a were associated with milder disease severity. Decreased miRNA-125a expression suggests impaired macrophage activation and suppressed NF-κB signaling, potentially preventing excessive immune responses. The role of reduced miRNA-429 expression in RSV infection and immune function remains unclear and warrants further investigation [56].

A prospective, multicenter study conducted in 2024 provided new insights into the role of miRNAs in the pathogenesis of RSV bronchiolitis. The study included 493 infants with confirmed infection. Clinical data (demographic, medical, environmental) were obtained from the subjects, and nasopharyngeal samples were collected for further analysis. The study assessed disease severity, defined by the need for respiratory support (oxygen therapy, CPAP, mechanical ventilation). Numerous connections between miRNAs, mRNA, and disease severity have been identified. Notably, miRNA-27a-3p was linked to disrupting microtubule anchoring in ciliated cells, leading to excessive damage. Let-7 family miRNAs affected LIM domain proteins involved in cellular junctions and epithelial barrier integrity. Additionally, miRNA-26b-5p, interacting with EBNA1BP2 (Epstein–Barr Nuclear Antigen 1 Binding Protein 2), influenced RNA processing, potentially altering mechanisms such as alternative splicing, thereby regulating innate immune responses and disease severity [57].

Recent studies have highlighted the variable expression of miRNAs in adult patients infected with RSV. Although this virus is primarily considered a threat to infants and older people, there is growing awareness of its significant impact on those with comorbidities. The increase in infections in high-risk patients, such as those with chronic lung disease or transplant recipients, highlights the need for further research, particularly into the role of miRNAs in the immune response. Profiling these molecules can aid in diagnosing and developing new therapies [58,59]. Studies of RSV vaccines have shown that differences in miRNA expression can help assess their safety and efficacy. miRNAs such as let-7 and miR-106a regulate inflammatory responses and immune function, and their further analysis may support the development of effective therapeutic and preventive strategies [60].

## 5. Influenza Virus

The influenza virus is a significant cause of respiratory infections in children, with types A and B responsible for seasonal epidemics. The clinical symptoms of influenza are non-specific, affecting both upper and lower respiratory tracts. Children, particularly those in primary school, are crucial in transmitting the virus within communities. Fortunately, the availability of annual vaccinations serves as an effective means of protection against seasonal illness, which is vital for controlling the spread of the disease [61,62].

Like previously discussed pathogens, the influenza virus can exacerbate asthma by remodeling the airways. A study on three specific miRNAs—miRNA-20a, miRNA-22, and miRNA-132 revealed intriguing results. Samples from the airway were obtained by bronchoscopy. The study group comprised adult patients with severe asthma; the control group comprised healthy volunteers. Then, bronchial epithelial cells were exposed to the influenza virus. Before infection, the levels of the molecules were similar. Still, after exposure to Air–Liquid Interface (ALI) cultures, miRNA-22 levels increased in healthy participants, inhibiting the expression of CD147 (Cluster of Differentiation 147), HDAC4 (Histone Deacetylase 4), and MMP-9 (Matrix Metalloproteinase-9). In contrast, in asthmatic cells, CD147 expression rose while miRNA-22 and HDAC4 levels dropped, linking miRNA-22 variations to asthma pathogenesis. For patients experiencing asthma exacerbations during influenza infections, developing a treatment approach focused on targeting miRNA-22 could result in severe disease complications [63].

MiRNAs play a crucial role in regulating immune responses and antiviral mechanisms by modulating the expression of genes involved in pathogen recognition and the inflammatory response. Influenza virus infections significantly change miRNA expression profiles, particularly influencing antiviral responses in lung epithelial cells. One such molecule is miRNA-144, which is crucial for TRAF regulation. Research shows that lower levels of miRNA-144 correspond to higher viral loads during early viral replication stages [64]. Additionally, miRNA-193b-5p is significant in H1N1 influenza virus infection (influenza A virus with Hemagglutinin type 1 and Neuraminidase type 1) induced by interferon beta. This miRNA is vital for maintaining intercellular junction integrity through occludin regulation. Elevated levels can disrupt this balance, but inhibiting miRNA-193b-5p enhances antiviral responses, reduces lung damage, and improves survival rates in infected models. Notably, its experimental delivery has shown promise in alleviating viral-induced damage [65]. In addition to the multiple functions of miRNA molecules, they are crucial for lung regeneration following viral infection. A distinct molecular mechanism of action has been demonstrated for miRNA-21. In mice, pneumonia was induced by exposure to the influenza virus. miRNA-21 expression was shown to increase following infection, as well as its effects on modulation of the inflammatory response and cell regeneration by controlling pro-inflammatory pathways and inhibiting NF-κB activity. Experimental inhibition of this molecule resulted in more severe morbidity. Conversely, the inhibition of miRNA-99a can lead to moderate health deterioration due to inflammatory response disruptions, while reduced levels of miRNA-145 appear less impactful on clinical symptoms and type II cell proliferation [66]. miRNA-4776 is another molecule shown to affect the NF-κB pathway by targeting the NF-κB beta inhibitor. Its increased levels during influenza virus infection indicate a protective mechanism against the spread of disease by regulating inflammatory responses [67]. After exposure to influenza A, miRNA-let7 levels decrease, but using a mimic increases type I interferon production, limiting viral spread [68].

Other essential functions of miRNAs in viral infections include their influence on pathogen replication and evasion of host immunity mechanisms. One such molecule is miRNA-200b-3p. It degrades TBK1 (TANK-binding kinase 1) mRNA via cAMP response element binding protein (CREB) and inhibits interferon production, allowing the virus to evade the immune system effectively [69]. Following exposure to a group B virus, miRNA-30e-3p levels rise, directly influencing the synthesis of hemagglutinin (HA) and neuraminidase (NA), thereby inhibiting infection spread [70]. MiRNA-205-5p also inhibits nucleoprotein (NP) production, which encases viral nucleic acid, providing a protective role. Interestingly, studies show that oseltamivir treatment increases miRNA-205-5p, suggesting an additional antiviral mechanism [71]. During infections with three subtypes of influenza A virus (pH1N1, H5N1, and H3N2), miRNA-3145 targets PB1 (Polymerase Basic Protein 1), a crucial component of the viral polymerase complex. Using a mimetic for this miRNA effectively inhibits replication across all virus subtypes. These findings underscore the potential for targeting specific miRNAs in therapeutic strategies against influenza infections [72].

Several studies have highlighted the critical role of miRNAs in modulating cellular responses during influenza infections, revealing complex interaction mechanisms between host miRNAs and viral processes. In one study, four miRNAs, including miRNA-141, miRNA-200c, miRNA-21-3p, and miRNA-29b-1-5p, were identified whose expression levels significantly changed in cells after exposure to two strains of the influenza A virus. Interestingly, reduced levels of miRNA-21-3p resulted in increased expression of HDAC8 (histone deacetylase 8), which protects the body from infection by limiting pathogen replication [73].

In 2023, research further emphasized miRNA-141′s role in influenza A, showing its inhibitory effect on antiviral proteins like MxA (Myxovirus Resistance Protein A) and STAT3 (Signal Transducer and Activator of Transcription) [74]. Additionally, miR-193b emerged as a key regulator, inhibiting influenza A replication by downregulating the Wnt/β-catenin signaling pathway and halting the cell cycle at the G0/G1 phase, effectively reducing viral loads and mitigating weight loss in animal models [75].

Comparative studies between influenza A and B infections revealed distinct miRNA profiles. For instance, miRNA-1260 and miRNA-374b-5p were significantly elevated in influenza A patients, while miRNA-205-5p was more prominent in influenza B cases. Remarkably, miRNA-374b-5p modulates the PI3K/AKT pathway, affecting viral replication, cell transition, and apoptosis. Moreover, associations between miRNAs—such as miRNA-326, miRNA-15b, miRNA-122, miRNA-885, miRNA-133a, and miRNA-150—and cytokines like IL-6, IL-15, and IL-17 suggest their potential as biomarkers for early influenza detection and strain differentiation [76].

Laboratory experiments further elucidated the role of miRNAs in influenza infections. Influenza A virus was shown to induce miRNA-1290 in human alveolar adenocarcinoma cells via the Extracellular Signal-Regulated Kinase pathway. Elevated miRNA-1290 inhibited vimentin expression, prolonging the retention of the viral ribonucleoprotein (vRNP) subunit PB2 in the nucleus. This enhancement of viral replication facilitated the production of new pathogens, underscoring the intricate relationship between miRNA dynamics and viral propagation [77].

## 6. Metapneumovirus (MPV)

Metapneumovirus is another common etiological factor causing acute respiratory infections, which are hazardous among the youngest and oldest patients. It has been shown that, like RSV, it shares a similar seasonality and clinical symptoms and is a leading cause of lower respiratory tract infections in about 40–60% of children. Like the previously mentioned pathogens, this virus also induces changes in miRNA expression in host cells [78,79].

An intriguing relationship was observed between RSV and hMPV (human MPV) in cell line infections. Both caused the expression of miRNA let-7f but with different effects. In hMPV, let-7f limited the virus’s replication, while in RSV, it did not. This underscores the variability of miRNA’s role in viral replication, depending on the pathogen. Additionally, the M2-2 protein in hMPV, which regulates viral RNA synthesis, suppressed miRNA-30 and miRNA-16, potentially aiding the virus in spreading more effectively by reducing the host’s antiviral defenses [80]. RSV and MPV were also at the center of interest in the following experiment. The studies consisted of culturing the viruses in appropriate cells and their purification. Dendritic cells derived from human monocytes were isolated from the blood of healthy donors and then infected with viruses. After 24 h, RNA was collected for miRNA extraction. The results showed different expressions of miRNA molecules. RSV increases the expression of miRNA-4448 and miRNA-30a-5p, which may affect the regulation of genes related to cell survival, dendritic differentiation, and autophagy. MPV, on the other hand, induces the production of miRNA-182-5p and miRNA-4634. Significant expression of the latter molecule was demonstrated. Possible molecular targets of miRNA-4634 during infection may include genes such as cyclin-dependent kinase inhibitor 2A (*CDKN2A*) and Toll-like receptor 8 (*TLR8*). These genes recognize viral and bacterial pathogens and regulate the cell cycle, suggesting that miRNA-4634 may influence the immune response and cell proliferation during infection [81,82,83].

The 2023 experiment again highlighted the importance of miRNA-4634. After macrophages were exposed to metapneumovirus, increased production of this molecule was observed. This simultaneously attenuated interferon function and allowed the virus to evade host antiviral defenses. Reduced IFN activity in epithelial cells also led to increased viral replication. This study highlights a strong connection between viral infections, IFN response, and miRNA expression, which influences the severity and progression of viral diseases [84].

MPV infection significantly elevates miRNA-182-5p levels in cells, disrupting the balance of Th17 cells and promoting a shift towards a Th2-type immune response. This shift leads to the release of Th2 cytokines like IL-13 and IL-5, triggering an increased recruitment of neutrophils to the lungs. The accumulation of neutrophils, combined with heightened cytokine levels, exacerbates inflammation in the respiratory system. This process intensifies the inflammatory response, contributing to more severe respiratory symptoms [85].

Studies on metapneumovirus infections demonstrate that the virus influences miRNA regulation, weakening the immune response and intensifying inflammation. In individuals with asthma, this can lead to more severe exacerbations, as such infections further compromise respiratory function. These findings highlight the critical role of miRNA in asthma development and underscore the importance of considering these mechanisms in treating asthmatic patients.

## 7. Severe Acute Respiratory Syndrome Coronavirus 2 (SARS-CoV-2)

At the end of 2019, unusual cases of pneumonia were observed among hospitalized patients in Wuhan, China. The coronavirus, SARS-CoV-2, was identified as the cause of the COVID-19 (Coronavirus Disease 2019) pandemic. This pathogen, which triggers acute respiratory distress syndrome, rapidly spread across the world, leading to a global health crisis. The symptoms exhibited by infected patients are non-specific. These symptoms include fever, cough, shortness of breath, difficulty breathing, diarrhea, vomiting, headache, and fatigue. Experiments have demonstrated that this pathogen enters cells using its spike protein, binding to receptors such as the angiotensin-converting enzyme 2 (ACE2). Interestingly, a higher susceptibility to severe COVID-19 has been observed among patients with non-allergic asthma. This may be due to an intensified baseline inflammatory state caused by other comorbidities, such as obesity or older age among those studied. These coexisting conditions may have a more significant impact on worsening the course of COVID-19 than asthma itself. Additionally, the lower frequency of hospitalizations among patients with atopic asthma was associated with reduced expression of the angiotensin-converting enzyme 2 receptor in atopy, which correlated with allergic sensitization, total IgE levels, and type 2 cytokines [17,86].

A significantly increased expression of miRNA-2392 was revealed among COVID-19 patients compared to healthy individuals. Moreover, the level of this miRNA increased proportionally with viremia. Its impact on various cellular and molecular mechanisms has been confirmed. miRNA-2392 plays a role in suppressing mitochondrial gene expression, leading to increased inflammation, glycolysis, and hypoxia, contributing to many infection symptoms [87]. Another molecule associated with SARS-CoV-2 infection is miRNA-150-5p. In an observational, prospective study, patients with confirmed infection participated. They were divided into three groups based on the severity of the disease: moderate, severe, and critical. Demographic, biochemical, and radiological data of the patients were collected. Blood samples were taken at three points: upon admission, day 7, and day 21 of hospitalization. Its decreased expression has been demonstrated in the serum of patients with moderate and severe forms of the disease.

The use of a mimetic effectively inhibited SARS-CoV-2 infection in vitro. The action of this molecule may be related to the regulation of the viral non-structural protein 10 (nsp10), which is involved in evading the host’s immune response and efficient replication. Together with other viral proteins (nsp16 and nsp14), it plays a role in concealing viral RNA, making it more difficult for host cells to detect the infection, and it also participates in RNA repair, increasing its stability and minimizing the risk of damage during replication. Through these functions, nsp10 plays an essential role in the progression of COVID-19 infection [88]. The concentration of miRNA-155-5p was associated with the severity of the disease and negatively correlated with the level of SOCS1 (suppressor of cytokine signaling 1). This protein influences the modulation of the immune response. This condition leads to increased cytokine synthesis and heightened inflammation. Another study demonstrated that miRNA-155 could effectively predict the risk of death from COVID-19 [89,90].

Previous discussions addressed the role of miRNA-146 in the context of viral infections. In the case of SARS-CoV-2 infection, an increased expression of miRNA-146a-5p was observed, exhibiting anti-inflammatory properties, in contrast to miRNA-146a-3p, which has pro-inflammatory characteristics. Furthermore, the first molecule dominates over the second, inhibiting its expression, indicating the significant role of the 5p form in modulating the inflammatory response in respiratory system cells. The molecules had an opposite effect on IL-8 levels. Clinical data analysis revealed that miRNA-146a-5p is upregulated during the acute phase of COVID-19. At the same time, miRNA-146a-3p is more strongly associated with chronic inflammatory conditions, suggesting their potential application as biomarkers in diagnosing and monitoring lung diseases [91].

In a subsequent study focusing on a pediatric population with confirmed SARS-CoV-2 infection, patients were divided into mild and severe disease groups, with severe cases being characterized by respiratory failure, shock, or death. Saliva samples were analyzed, particularly concerning the expression of four microRNAs (miRNA-296-5p, miRNA-4495, miRNA-548ao-3p, and miRNA-1273c). It was found that the levels of these molecules were reduced, and the lower their expression, the more severe the disease. These microRNAs are involved in critical processes such as viral protein processing, virion assembly, immune system function, homeostasis, tissue repair, and phagocytosis. Reduced expression of these molecules may weaken the body’s defense mechanisms during infection. SARS-CoV-2 RNA can “capture” host miRNA molecules, disrupting their function [92].

The study also showed that the reduction of miRNA-296 inhibited the growth of the 293T cell line. Conversely, a decrease in miRNA-602 led to faster development of HUVEC (Human Umbilical Vein Endothelial Cells) and increased levels of inflammatory proteins such as IL-6 and TNF-α. These disruptions affect normal cellular functions and may lead to changes in immune and inflammatory responses within the body [93]. Moreover, the genetic material of SARS-CoV-2, as an exogenous competitive RNA, may lead to an increased concentration of miRNA-1207-5p. In the context of severe COVID-19, this molecule influences the *CSF1* gene (colony-stimulating factor 1 or macrophage colony-stimulating factor), which may result in increased recruitment of macrophages and an intensified inflammatory response [94]. Further research suggests that host miRNAs, including miRNA-17-5p, miRNA-20b-5p, and miRNA-323a-5p, may exhibit antiviral activity against COVID-19 by specifically inhibiting various Toll-like receptor pathways, potentially modulating the host’s antiviral defense mechanisms [95].

## 8. Conclusions and Future Research Directions

This review examines how infections alter miRNA expression and explores their role in exacerbating asthma symptoms during infections. Asthma has been shown to influence miRNA expression profiles, modifying the host’s immune response to pathogens. Simultaneously, pathogens can manipulate host miRNAs to enhance their survival and replication. These molecules play a dual role by combating viruses and modulating the course of infections, adding complexity to the interplay between asthma and viral pathogens. This publication is one of the first to examine miRNA expression variability, specifically in pediatric asthma exacerbations caused by common viral infections, highlighting its novel contribution to the field. This publication is one of the first to examine miRNA expression variability, specifically in pediatric asthma exacerbations caused by common viral infections, highlighting its novel contribution to the field.

Airway obstruction, chronic inflammation, and the “asthmatic phenotype” are characterized by a chronic inflammatory response that modifies the immune system to be more aggressive toward pathogens. That leads to asthma becoming even worse. Such a paradigm of asthma exacerbation sets in and leaves the body in such a state that it cannot fight off any infections. There are also changes in miRNA expression patterns in asthma patients with associated infections. These changes can be used as disease markers that track progression, and they can be used as targets for drug interventions. Focusing on miRNAs could shift the focus of treatment from drug therapies that essentially treat the symptoms to actual recessive molecular mechanisms, which would usher in more scientists considering the various causes and treating asthmatics, reducing exacerbations, reducing the effects, and decreasing hospital stays. Nevertheless, the absence of well-designed randomized controlled trials emphasizes the importance of additional investigation into efficacy and safety.

The role of miRNAs in modulating inflammatory responses extends beyond asthma and has been implicated in various viral infections, including SARS-CoV-2, HCV, and HSV. In SARS-CoV-2, miRNAs influence the inflammatory response and disease progression, with molecules like miRNA-146a-5p exhibiting anti-inflammatory effects. Similar functions are observed in other viruses, such as miRNA-122 in HCV [96], which facilitates viral replication, and miRNA-146a in HSV [97], which modulates the inflammatory response. While these mechanisms share commonalities, the specific roles of miRNAs often vary depending on the virus, highlighting the need for further investigation into their diverse functions.

Future research should focus on characterizing miRNA expression profiles specific to different viruses and stages of infection. This could identify reliable biomarkers for diagnosis and disease monitoring while informing the development of miRNA-modulating therapies. Such therapies hold promise for broad applications in treating viral infections, inflammatory diseases, and conditions like asthma. Advancing our understanding of miRNAs in immune responses may pave the way for innovative treatments and preventative strategies in clinical practice.

## Figures and Tables

**Figure 1 ijms-26-00120-f001:**
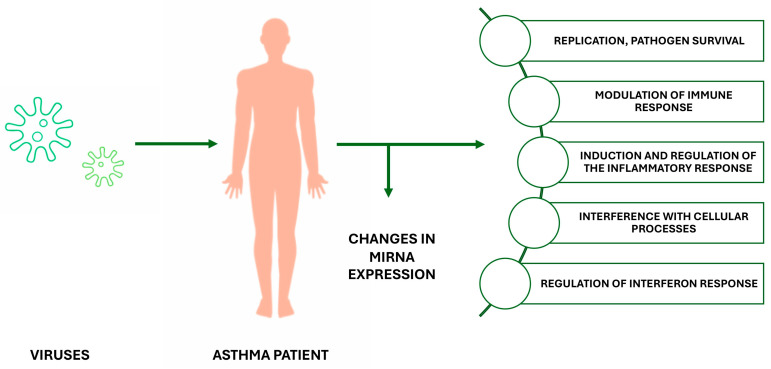
Functions of miRNA molecules induced during viral infection in patients with asthma.

## Data Availability

No new data were created or analyzed in this study. Data sharing is not applicable to this article.

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
