# Peer review of "The Molecular Basis of Asthma Exacerbations Triggered by Viral Infections: The Role of Specific miRNAs"

_ijms, 2024, doi:10.3390/ijms26010120_

Round 1
Reviewer 1 Report
Comments and Suggestions for Authors
Thank you for the opportunity to review the article entitled “The Molecular Basis of Asthma Exacerbations Triggered by Viral Infections: The Role of Specific miRNAs”. This is a very well written review article presenting changes in the expression of miRNAs during the five most common viral infections (RV, RSV, Influenza virus, MPV and SARS-CoV-2) causing asthma worsening, with particular emphasis on the pediatric population. Authors present that asthma can impact expression profiles and modify how the host responds to infections, as well as that future studies could hopefully help us identify patients with specific immune responses towards a precision medicine approach. This manuscript covers a very hot topic in the field and is an important addition to literature.
Some minor comments follow:
1. Based on recent studies with regards to RSV infections in adulthood not only on the pediatric population, it should be further discussed that alterations in miRNA could potentially be expressed in adult patients, as well. These recently published articles which summarize current knowledge for burden of respiratory syncytial virus infection in adults could be helpful:
Burden of respiratory syncytial virus infection in older and high-risk adults: a systematic review and meta-analysis of the evidence from developed countries. Eur Respir Rev. 2022 Nov 15;31(166):220105.
Respiratory syncytial virus in adults with comorbidities: an update on epidemiology, vaccines, and treatments. Clin Microbiol Infect. 2023 Dec;29(12):1538-1550.
2. Given the lack of RCTs, authors could add a short comment with regards to the need for RCTs with endotyping in the discussion section.
Author Response
Dear Sir or Madam,
We are grateful for your review and appreciate our article's valuable comments. We have revised our manuscript, according to the comments and suggestions point by point, believing that the manuscript has been further improved and would match your expectations.
- I have included additional information regarding potential changes in miRNA expression in adult patients with RSV infection to extend the analysis beyond the pediatric population.
Recent studies have highlighted the variable expression of miRNAs in adult patients infected with RSV. Although this virus is primarily considered a threat to infants and older people, there is growing awareness of its significant impact on those with comorbidities. The increase in infections in high-risk patients, such as those with chronic lung disease or transplant recipients, highlights the need for further research, particularly into the role of miRNAs in the immune response. Profiling these molecules can aid in diagnosing and developing new therapies [58,59]. Studies of RSV vaccines have shown that differences in miRNA expression can help assess their safety and efficacy. miRNAs such as let-7 and miR-106a regulate inflammatory responses and immune function, and their further analysis may support the development of effective therapeutic and preventive strategies [60].
- In the discussion section, I commented on the necessity of conducting RCTs with endotyping, which aligns with the suggested suggestion.
Nevertheless, the absence of well-designed randomized controlled trials emphasizes the importance of additional investigation into efficacy and safety.
Yours sincerely,
Natalia Kierbiedź- Guzik
Reviewer 2 Report
Comments and Suggestions for Authors
This review described the roles of miRNAs in the asthma exacerbations triggered by viral infections. The topic was meaningful and suitable. I recommend a major revision on the following points:
1. In the introduction, the main novelty of this review should be clearly provided. The present version was not distinguished from the similar reviews.
2. In the last paragraph of the second section, the authors should demonstrate the organization logic of the next paragraphs. Were they appeared in the order of the spread of infection, fatality, or chronological order?
3. The inner logic in each section of the viruses should also be arranged. If the investigations were extensive, the authors should tell the story with the progress.
4. Although SARS-CoV2 was included, the authors should add another discussion section to compare the point views in this review with the similar ones, and to lead the orientation of future research.
5. The language use should be improved.
Comments on the Quality of English Language
Moderate review needed.
Author Response
Dear Sir or Madam,
We would like to express our sincere gratitude for your review and the invaluable comments provided on our manuscript. We have made corrections to the article according to the comments. Your feedback has been instrumental in enhancing the quality of our work, and we truly appreciate the time and effort you dedicated to evaluating our submission.
- The main novelty of this review has been explicitly highlighted in the conclusion section, where we emphasized its unique focus on the interplay between miRNA expression, viral infections, and asthma exacerbations. This distinct perspective sets it apart from similar reviews.
This publication is one of the first to examine miRNA expression variability, specifically in pediatric asthma exacerbations caused by common viral infections, highlighting its novel contribution to the field.
- The pathogens discussed in the subsequent paragraphs have been organized based on their frequency of causing asthma exacerbations in children, providing a logical flow that aligns with their clinical significance.
The pathogens in the paragraph are listed in order of those most frequently causing asthma exacerbations in children, concluding with SARS-CoV-2, a novel pathogen that may also significantly worsen the disease.
- The sections discussing the viruses have been reorganized to ensure a more precise and logical structure. The narrative now follows the progression of research findings, highlighting key developments and advancements in understanding the role of miRNAs in each viral infection.
- As suggested, an additional discussion section has been incorporated to compare the perspectives presented in this review with those from similar studies. Furthermore, the content from the "Conclusion" and "Future Research Directions" sections has been integrated to provide a cohesive narrative. This revised structure outlines the key findings, contextualizes them within the broader research landscape, and highlights future research orientations effectively.
- The language throughout the manuscript has been revised and improved to enhance clarity, readability, and academic tone.
Yours sinceraly,
Natalia Kierbiedź- Guzik
Round 2
Reviewer 2 Report
Comments and Suggestions for Authors
The authors have revised the submission rationally. I recommned the acceptance.